# Spontaneous symmetry breaking in free theories with boundary potentials

**Vladimir Procházka⋆ and Alexander Söderberg†**

Department of Physics and Astronomy, Uppsala University,
Box 516, SE-751 20 Uppsala, Sweden

⋆ vladimir.prochazka@physics.uu.se, † alexander.soderberg@physics.uu.se

## Abstract

Patterns of symmetry breaking induced by potentials at the boundary of free $O(N)$-models in $d = 3 - \epsilon$ dimensions are studied. We show that the spontaneous symmetry breaking in these theories leads to a boundary RG flow ending with $N-1$ Neumann modes in the IR. The possibility of fluctuation-induced symmetry breaking is examined and we derive a general formula for computing one-loop effective potentials at the boundary. Using the $\epsilon$-expansion we test these ideas in an $O(N) \oplus O(N)$-model with boundary interactions. We determine the RG flow diagram of this theory and find that it has an IR-stable critical point satisfying conformal boundary conditions. The leading correction to the effective potential is computed and we argue the existence of a phase boundary separating the region flowing to the symmetric fixed point from the region flowing to a symmetry-broken phase with a combination of Neumann and Dirchlet boundary conditions.

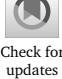
# 1 Introduction

Spontaneous breaking of global symmetries is one of the most universally used tools to understand phase transitions in modern theoretical physics. In this paper we would like to consider its application to systems described by scalar field theories existing on a manifold with a boundary. A lot has already been understood in the condensed matter context [1], where such systems describe polymer absorption by walls [2]. Other than the usual order-disorder phase transition in the bulk (called the ordinary transition), there is a possibility of an extraordinary phase transition at the boundary above the bulk critical temperature. Field theoretically such systems are represented by an $O(N)$-model in $d = 3$ dimensions with polynomial interactions in the bulk where the extraordinary phase transition is triggered by a negative 'boundary mass' term. This representation makes them amenable to study with the techniques of high-energy physics. In particular the machinery of boundary conformal bootstrap [3] allows for high precision evaluation of correlation functions at the Wilson-Fisher (WF) fixed point (f.p.) [4, 5], which was recently used in evaluation of layer susceptibility at the extraordinary transition point [6, 7]. Alternatively a wealth of information on these models can be obtained by coupling them to a curved background and calculating the resulting partition function [8, 9].

In this work we would like to examine the case when the bulk couplings are turned off and instead we include interactions at the boundary. For $d = 3 - \epsilon$ this still leads to a non-trivial RG flow at the boundary with an interacting infrared (IR) f.p., which was recently studied in [10] and [11]. Scalar models with boundary interactions were considered long before in condensed matter literature [12]. In the context of polymer physics, tuning the bulk couplings to zero means considering a rather non-realistic example with two-body monomer interactions confined to the boundary.

In the realm of high energy physics there are nevertheless important examples of free theories with boundary interactions. For $d = 2$ free bosons with boundary potentials have been studied in the context of open strings [13, 14]. More recently there has been a progress in constraining free scalar theories with boundaries and defects with $d > 2$ by using conformal bootstrap techniques [15, 16].

Finally let us note that free models are often related to interacting ones via dualities such as bosonisation in $d = 2$ or more refined dualities that have recently been discovered in $d = 3$ [17, 18]. Thus it is possible that already by studying the models that are free in the bulk we can learn something about the interacting theories and their boundary deformations via the duality.

In this paper we would like to consider giving a vacuum expectation value (v.e.v.) to a boundary field. This is not a new idea in itself, e.g. in the condensed matter context (cf. [1]) this phenomenon gives rise to new kinds of phase transitions called the special and the extraordinary. These transitions cannot be deduced from the knowledge of the bulk phase diagram itself and are described by a set of independent boundary parameters (couplings, v.e.v.'s, etc.). When the bulk is free there are no bulk parameters to tune so all the non-trivial dynamics happens at the boundary either through edge degrees of freedom or dynamical boundary conditions (b.c.'s). We would like to study the latter in the present work and convince the reader that such a simple set-up can lead to rich physics similar to the phase structure of the Ising model.

Let us start by introducing the class of models we want to work with. We will consider a free $O(N)$-scalar with a boundary potential

$$S[\phi] = \int_{\mathbb{R}^d_+} d^d x \left( \frac{(\partial_\mu \phi)^2}{2} + \delta(x_\perp) V(\phi) \right), \quad d = 3 - \epsilon, \tag{1}$$

where $\mathbb{R}^d_+ = \{(x_\parallel, x_\perp) : x_\parallel \in \mathbb{R}^{d-1}, x_\perp > 0\}$, we have suppressed the index notation for $\phi \equiv \phi^i$

with $i$ running from 1 to $N$ and used the Euclidean space conventions. The bulk theory has an $O(N)$-symmetry

$$\phi \to R\phi, \quad R \in O(N), \tag{2}$$

and an additional shift symmetry

$$\phi \to \phi + c. \tag{3}$$

Here $c$ is a constant vector.[1] In the absence of boundary potential we can choose Neumann b.c.'s, which will preserve both of these symmetries.

The boundary potential will break the bulk shift symmetry, but we will assume that it preserves the $O(N)$-symmetry. We have only chosen $O(N)$ for simplicity but all of the following discussion can be generalized to other compact global symmetry groups. The equations of motion (e.o.m.) together with the b.c.'s describing the system in (1) read

$$\partial^2 \phi = 0, \quad \partial_\perp \phi|_{x_\perp = 0} = V'(\phi)|_{x_\perp = 0}. \tag{4}$$

If the potential has any non-trivial minima these equations admit a constant solution $\phi = \langle \phi \rangle \neq 0$ satisfying

$$\frac{\partial V}{\partial \phi^i}(\langle \phi \rangle) = 0. \tag{5}$$

We will furthermore assume that the solution is a stable minimum with $\frac{\partial^2 V}{\partial \phi^i \partial \phi^j} \geq 0$ (by this we mean that the Hessian matrix has only non-negative eigenvalues). Now what are the consequences of having such solution? The vacuum $\langle \phi \rangle$ will break the global $O(N)$-symmetry down to $O(N-1)$. Had there been no boundary interaction this would obviously not be the case since the new vacuum would be related to the trivial one by the shift symmetry. We will now demonstrate that in the presence of a boundary the expansion around $\langle \phi \rangle$ leads to a distinct qualitative picture.

By running the usual textbook arguments leading to the Goldstone theorem we see that the Hessian matrix $\frac{\partial^2 V}{\partial \phi^i \partial \phi^j}$ has exactly $N-1$ vanishing eigenvalues corresponding to the broken generators of $O(N)$. We can choose the usual parametrisation to expand about the minimum

$$\phi = e^{\eta^k T^k}(\langle \phi \rangle + \sigma). \tag{6}$$

Here $T^k$, $k \in \{1,\ldots,N-1\}$ is the generator of the Lie algebra corresponding to $O(N)/O(N-1)$ and $\sigma$ is a vector in the flavour space parallel to $\langle \phi \rangle$ satisfying $|\sigma| \ll |\langle \phi \rangle|$. If we insert (6) into the potential (1) we find that $\eta^k$ is a free massless field and that $\sigma$ has a positive boundary mass and both cubic as well as quartic interactions[2]

$$S[\eta, \sigma] = \int_{\mathbb{R}^d_+} d^d x \left[ \frac{(\partial_\mu \eta^k)^2}{2} + \frac{(\partial_\mu \sigma)^2}{2} + \delta(x_\perp) V(\sigma) \right] + \ldots,$$
$$V(\eta, \sigma) = \frac{m}{2} \sigma^2 + \mathcal{O}(\sigma^3), \tag{7}$$

where $m > 0$ corresponds to the nonzero eigenvalue of $\frac{\partial^2 V}{\partial \phi^i \partial \phi^j}$. This mass term induces a boundary RG flow for $\sigma$ into Dirichlet b.c. in the IR.[3] The fields $\eta^k$ are similar to the usual

---

[1]For a compact scalar $\phi$ in three dimensions the symmetry can be interpreted as a topological $U(1)$ that acts on the corresponding magnetic monopoles $e^{i\phi}$. For bosonic strings on a worldsheet this symmetry corresponds to space-time translations.

[2]Here we used that $e^{\eta^k T^k} \in O(N)/O(N-1) \subset O(N)$, which means that $\phi^2 = (\langle \phi \rangle + \sigma)^2$.

[3]By IR we mean large distances parallel to the boundary.

Goldstone bosons in that they gain no boundary potential and therefore will retain the Neumann b.c.'s in the IR. This gives us a clear picture of how the symmetry breaking is realized in the IR: the flow will leave us with $N-1$ free Neumann scalars preserving the $O(N-1)$- and the shift-symmetry. The remaining field satisfies Dirichlet b.c. and therefore its boundary propagator vanishes. This is similar to the tachyon condensation in open string theory [14] with the preserved $O(N-1)$- and shift-symmetry being the rotations and translations preserving the IR D-brane. Note that since the symmetry breaking only affects the b.c.'s the effect gets weaker as we depart from the boundary. More concretely by examining the propagators corresponding to (7) at large perpendicular distance[4] (cf. (63)) we find that both $\eta^k$ and $\sigma$ have the same asymptotics implying the preserved $O(N)$-symmetry deep in the bulk as expected.

In a quantum theory the constant solution to (4) can only exist in the absence of bulk couplings. Were there any bulk couplings the solution to the e.o.m. would acquire a dependence on the normal coordinate and we would need to deal with the renormalisation of $\phi$ in the near boundary limit.[5] As a consequence the v.e.v.'s of bulk and boundary fields become unequal, which leads to so called extraordinary phase transitions (see [1] for a comprehensive review of phase transitions with boundaries).

In the case of a free bulk that we consider here, the v.e.v. of a bulk field $\phi$ is completely determined from the boundary potential $V$. This is in line with the fact that in the absence of bulk interactions, $\phi$ does not renormalise at the boundary (i.e. $\lim_{x_\perp \to 0} \phi = \hat{\phi}$ is well defined).[6] Thus to understand the IR dynamics of such fields theories we need to determine the potential at the quantum level.

For a potential without non-zero local minima we have two possibilities. Either there exists a boundary RG flow into an IR f.p. satisfying conformal b.c.'s[7] or new minima appear through quantum corrections. The former scenario is analogous to second-order phase transitions in statistical physics as it involves an IR f.p. with calculable critical exponents (scaling dimensions of boundary operators). The latter corresponds to a fluctuation induced first-order phase transition with the order parameter $\langle \phi \rangle$. At the perturbative level the quantum corrections to the classical potential come from the loops through the Coleman-Weinberg (CW) mechanism [21].

Since the example we consider in this paper involves a boundary with low dimensionality ($\leq 2$) we should comment on the Mermin-Wagner theorem [22] prohibiting spontaneous symmetry breaking of continuous global symmetries in $d \leq 2$. For bulk dimension $d > 2$ this is not an issue due to the boundary theory being non-local.[8] The $d = 2$ case is more subtle due to the IR divergences of the free scalar propagator screening the v.e.v. and thus preventing the long range order. Nevertheless based on the discussion above we still expect to see the emergence of Dirichlet scalar described under (7) at large boundary distances. This is because the boundary primary operator corresponding to a Dirichlet scalar is actually $\partial_\perp \phi|_{x_\perp=0}$ which has a well defined IR behaviour. Nevertheless it is not clear to the authors whether one can still interpret this as a phase transition. We will return to this point at the end of the explicit example in section 3.2.

In section 2 we will show how to compute the quantum corrections to the potential at the one-loop level for theories of type (1). We illustrate how these ideas can be implemented

---

[4]In appendix A we find a general expression for the spacetime and the momentum propagator in a boundary quantum field theory (BQFT) with a mass term in both the bulk and on the boundary.

[5]By this we mean that the field enjoys the boundary operator expansion $\phi = x_\perp^{-\Delta+\hat{\Delta}} \hat{\phi} + \ldots$, where $\hat{\phi}$ is a boundary operator of dimension $\hat{\Delta}$. As shown in [10], this expansion is actually equivalent to operator renormalisation and $\hat{\phi}$ can be interpreted as renormalised field.

[6]See [10,11] and the earlier work [19] for a proof of this statement.

[7]The conformal b.c.'s of [20] imply vanishing of the normal-parallel components of the bulk energy-momentum at the boundary. It was shown in [10] that for models of the kind (1) this is equivalent to vanishing of the boundary $\beta$-functions.

[8]We would like to thank an anonymous referee for pointing this out.

in a scalar theory with $O(N) \oplus O(N)$-symmetry with interactions confined to the boundary in section 3. Finally in section 4, we discuss the broader picture and some future extensions of this work.

## 2 One loop effective boundary potentials

In the following we will assume the existence of a classical potential $V(\phi)$ at the boundary. For simplicity we will consider a single scalar field in the bulk, and later generalize this to $O(N)$. We will expand the action (1) with $\phi = \phi_{\mathrm{cl}} + \delta\phi$ about the classical minimum background $\phi_{\mathrm{cl}}$ satisfying the e.o.m. (4).[9] The linear terms vanish by virtue of the e.o.m. and we will only keep the quadratic part of the potential

$$V_{quad} = \frac{M}{2}(\delta\phi)^2 + \mathcal{O}\left(\delta\phi^3\right), \quad M = V''(\phi = \phi_{\mathrm{cl}}) > 0. \tag{8}$$

The bulk action for $\delta\phi$ will be the one of a free massless scalar. The one-loop effective potential will therefore be obtained by computing the functional determinant of the operator

$$D = -\partial^2, \tag{9}$$

subject to the following b.c.

$$\lim_{x_\perp \to 0}(\partial_\perp - M)\phi = 0. \tag{10}$$

In general a functional determinant of a differential operator $D$ is computed using

$$\det D = e^{-\frac{1}{2}\mathrm{tr}\log D}, \tag{11}$$

where the trace is evaluated in a suitable (complete) basis of functions $\{\phi_n\}$. I.e. we have

$$\mathrm{tr}\log D = \sum_n \int_{\mathbb{R}^d_+} d^d x \, \phi_n^* \log D \phi_n. \tag{12}$$

Without a boundary we typically take the complete set of eigenfunctions of $D$. For example in the case of $D = -\partial^2$ we take $\phi_n \to \phi_p = e^{ipx}$ and the sum over $n$ turns into a momentum space integral.

In our case we have to impose the b.c. (10) on the eigenfunctions. The corresponding functional determinant will take the form

$$\mathrm{tr}\log D = \int_{\mathbb{R}^d} \frac{d^d p}{(2\pi)^d} \int_{\mathbb{R}^d_+} d^d x \, \tilde{\phi}_p^* \log D \tilde{\phi}_p, \tag{13}$$

with the momentum eigenfunctions satisfying (10). More concretely they read

$$\tilde{\phi}_p(x) = \frac{1}{\sqrt{2}}\left(e^{ipx} + \frac{ip_\perp - M}{ip_\perp + M}e^{i\tilde{p}x}\right), \tag{14}$$

where we defined a reflected momentum $\tilde{p} = (p_\parallel, -p_\perp)$. By substituting these eigenfunctions in (14) we get

$$\mathrm{tr}\log D = \int_{\mathbb{R}^d} \frac{d^d p}{(2\pi)^d} \int_{\mathbb{R}^d_+} d^d x \left(1 - i\frac{-M + ip_\perp}{p_\perp - iM}e^{-2ip_\perp x_\perp}\right)\log(p^2). \tag{15}$$

---

[9]There is a factor of $\hbar = 1$ in front of the quantum fluctuations $\delta\phi$.

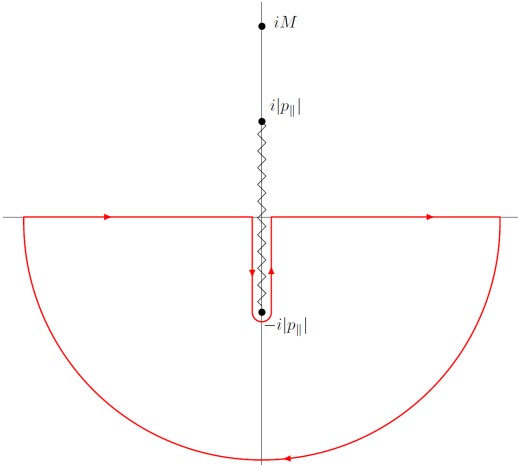

Figure 1: Integration countour for $M > |p_\parallel|$ closed in the lower plane with a branch cut between $(-i|p_\parallel|, +i|p_\parallel|)$ and a simple pole at $iM$. In the case $M < |p_\parallel|$ we take the branch cuts between infinity and $\pm i|p_\parallel|$ to move the pole away from them (this will not change the resulting integral).

The first term inside the bracket in (15) corresponds to the usual (IR divergent) bulk contribution. The second term is a new boundary contribution. We can evaluate it by first calculating the integral over $p_\perp$

$$\int_{\mathbb{R}} dp_\perp \left(-i\frac{M + ip_\perp}{p_\perp + iM}\right) [\log(|p_\parallel| + ip_\perp) + \log(|p_\parallel| - ip_\perp)] e^{-2ip_\perp x_\perp} . \tag{16}$$

This integral is evaluated by using the contour shown on figure 1. We close the contour in the lower half-plane so that the integral along the semicircle at infinity vanishes. This will also imply that the residue at $iM$ will not contribute. The integral (16) will therefore reduce to integrating the segment around the branch point at $-i|p_\parallel|$ which evaluates to

$$2\pi \int_0^{|p_\parallel|} du \frac{u - M}{u + M} e^{-2x_\perp u} . \tag{17}$$

This expression is still to be integrated over $x_\perp > 0$, which will turn (17) into

$$2\pi \frac{1}{2} \int_0^{|p_\parallel|} du \frac{u - M}{u(u + M)} . \tag{18}$$

The integral in the above expression can now be evaluated by standard methods

$$\int_0^{|p_\parallel|} du \frac{u - M}{u(u + M)} = -\log\left(\frac{|p_\parallel|}{\mu_{\text{IR}}}\right) + 2\log\left(\frac{|p_\parallel| + M}{M}\right), \tag{19}$$

where $\mu_{\text{IR}}$ is an IR cutoff introduced to regulate the IR divergence in the above integral.[10] Finally putting everything together we find the boundary contribution to the functional determinant (15)

$$\text{tr} \log D|_{\partial\mathcal{M}} = \int_{\mathbb{R}^{d-1}} d^2x_\parallel \int_{\mathbb{R}^{d-1}} \frac{d^{d-1}p_\parallel}{(2\pi)^{d-1}} \left[-\frac{1}{2}\log\left(\frac{|p_\parallel|}{\mu_{\text{IR}}}\right) + \log\left(\frac{|p_\parallel| + M}{M}\right)\right] . \tag{20}$$

---

[10]Physically this divergence arises from the infinite volume limit (or more specifically it comes from the $x_\perp \to \infty$ region of the original integral).

The first term in (20) does not depend on $M$ and therefore will not contribute to the effective potential. So we are left with the second term. From the path integral we have

$$-\frac{1}{2}\text{tr}\log D = V_{\text{eff}}^{\text{1-loop}} + \dots \tag{21}$$

Here the dots stand for derivative corrections. Thus we find that the non-trivial contribution to the boundary effective potential at one-loop reads

$$-\int_{\mathbb{R}^{d-1}} \frac{d^{d-1}p_{\parallel}}{(2\pi)^{d-1}} \log\left(\frac{|p_{\parallel}| + M}{M}\right). \tag{22}$$

Note that the denominator of the logarithm in (22) leads to a non-analytic power divergence $\Lambda^{d-1}\log M$. Such term does not appear in the usual bulk CW computation, but we can choose a suitable subtraction scheme to remove it,[11] so the relevant one-loop contribution to the effective potential reads

$$V_{\text{eff}}^{\text{1-loop}} = -\int_{\mathbb{R}^{d-1}} \frac{d^{d-1}p_{\parallel}}{(2\pi)^{d-1}} \log\left(|p_{\parallel}| + M\right). \tag{23}$$

For $N > 1$ this formula still holds with $M$ promoted to a matrix covariant under the global symmetry group. While the discussion so far has been focused on $O(N)$, the argument will hold for more general symmetry groups, where the explicit computation depends on the form of $M$. We will therefore proceed to evaluate the remaining integral over $(d-1)$-dimensional momenta in the next section.[12]

## 3 $O(N_{\phi}) \oplus O(N_{\chi})$ scalar model

### 3.1 The model

In this section we will consider an $O(N_{\phi}) \oplus O(N_{\chi})$ scalar model similar to that in [24–26], but with interactions happening at the boundary instead of in the bulk. The model will be defined by the following action[13]

$$S[\phi, \chi] = \int_{\mathbb{R}^d_+} d^d x \left(\frac{(\partial\phi)^2}{2} + \frac{(\partial\chi)^2}{2} + \delta(x_{\perp})V(\phi^2, \chi^2)\right),$$
$$V(\phi^2, \chi^2) = \frac{\lambda}{8}(\phi^2)^2 + \frac{\xi}{8}(\chi^2)^2 + \frac{g}{4}\phi^2\chi^2. \tag{24}$$

The scalar fields $\phi \equiv \phi^i$, $i \in \{1, \dots, N_{\phi}\}$ and $\chi \equiv \chi^a$, $a \in \{1, \dots, N_{\chi}\}$ satisfy $O(N_{\phi})$- as well as $O(N_{\chi})$-symmetry respectively. The potential in (24) breaks the $O(N_{\phi} + N_{\chi})$ bulk symmetry down to $O(N_{\phi}) \oplus O(N_{\chi})$. In general the potential can also include other relevant operators such as boundary masses[14] which we do not include as we do not want to have any symmetry breaking at the classical level (for a more detailed treatment of boundary masses we refer the interested reader to appendix A).

---

[11]More specifically this term will be set to zero by the CW renormalisation conditions (37) and addition of a 'cosmological' constant counterterm similar to (38). Addition of such counterterm does not change the form of the *renormalised* effective potential so we will not discuss it further.

[12]It should be noted that the relevant integral (23) was also computed in the condensed matter literature [23] for a diagonal $M$ in a different context (contribution to the Casimir free energy in the presence of Robin b.c.'s). The authors would like to thank H. W. Diehl for pointing this out.

[13]We consider scalar couplings, although in general they can be promoted to be tensorial [27].

[14]Other relevant operators involve the normal derivatives $\partial_{\perp}\phi^2$ and $\partial_{\perp}\chi^2$. However these are related to $\phi^2, \chi^2, \phi^4$ and $\chi^4$ through the e.o.m.

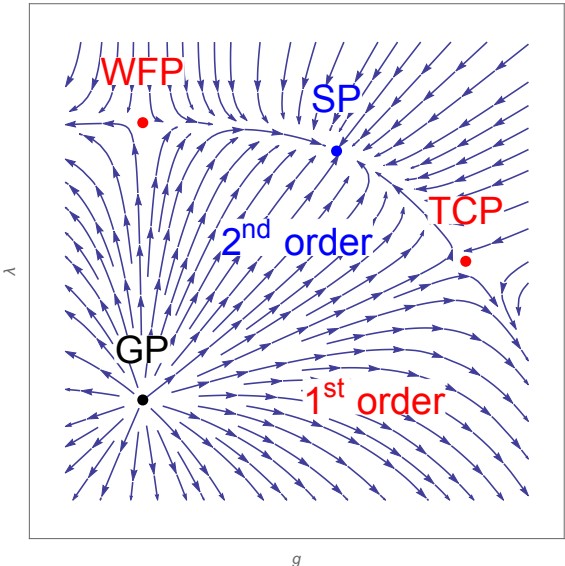

Figure 2: The RG flow for the model (24) when $N_\phi = N_\chi = 1$. F.p.'s are marked by dots, where the black dot is the fully repellent Gaussian f.p. (GP), the red dots (WFP and TCP) define a separatrix that separates regions corresponding to first- and second-order phase transitions and the blue dot is a fully attractive f.p. (SP) that is stable in the IR. The order parameter for the first order transition corresponds to $\langle \phi \rangle$.

To simplify the computations we take $N_\phi = N_\chi \equiv N$ and therefore also $\lambda = \xi$. This simplifies the classical potential down to

$$V(\phi^2, \chi^2) = \frac{\lambda}{8}(\phi^2)^2 + \frac{\lambda}{8}(\chi^2)^2 + \frac{g}{4}\phi^2\chi^2 = \frac{\lambda}{8}\left(\phi^2 + \chi^2\right)^2 + \frac{g-\lambda}{4}\phi^2\chi^2, \qquad (25)$$

where we have made the residual $O(2N)$-symmetry and the coupling that breaks it manifest.[15] In this case the theory also has an additional $\mathbb{Z}_2$ symmetry

$$\phi \longleftrightarrow \chi. \qquad (26)$$

From dimensional analysis we have the following engineering dimensions

$$\Delta_\phi = \Delta_\chi = \frac{d-2}{2} = \frac{1-\epsilon}{2}, \quad \Delta_\lambda = \Delta_\xi = \Delta_g = 3-d = \epsilon. \qquad (27)$$

A detailed discussion of the renormalisation of such models has been presented in our earlier work [10]. In appendix B we compute the $\beta$-functions for a model with generic $\lambda$, $\xi$ up to order two in the coupling constants. For $\xi = \lambda$ we have the following $\beta$-functions

$$\beta_\lambda = -\epsilon\lambda + \frac{N+8}{4\pi}\lambda^2 + \frac{N}{4\pi}g^2 + \dots, \quad \beta_g = -\epsilon g + \frac{g^2}{\pi} + 2\frac{N+2}{4\pi}\lambda g + \dots \qquad (28)$$

These $\beta$-functions have one Gaussian, and three WF f.p.'s defining a boundary RG flow chart depicted on figure 2. The positions of these f.p.'s read

$$(g^*, \lambda^*) \in \left\{ (0,0), \left(0, \frac{4\pi\epsilon}{N+8}\right), \left(\frac{2\pi(4-N)\epsilon}{N^2+8}, \frac{2\pi N\epsilon}{N^2+8}\right), \left(\frac{2\pi\epsilon}{N+4}, \frac{2\pi\epsilon}{N+4}\right) \right\}. \qquad (29)$$

---

[15]This splitting of the terms in the potential is not unique. A more detailed analysis along the lines of [27] would reveal that one should project the couplings on irreducible representations of $O(2N)$.

The first one is the fully repulsive Gaussian f.p. (GP), the second of these corresponds to decoupled $O(N)$-models with a single coupling at a WF point (WFP) studied in [10, 11], the third one (TCP) defines a stable solution only for $N < 4$, while the last f.p. (SP) enjoys an emergent $O(2N)$-symmetry. As already mentioned, the fundamental field $\phi$ does not acquire an anomalous dimension at these f.p.'s. On the other hand the composite operators (eg. $\phi^2$, $\chi^2$ etc.) have to be renormalised due to divergences in the boundary limit which leads to their anomalous dimensions in perturbation theory [10].

The flow diagram in figure 2 shares many features with the corresponding charts of the Abelian-Higgs model or the bulk $O(N) \oplus O(N)$-model (see for example [26]). In particular the diamond region corresponds to the domain of attraction of the symmetric, IR stable critical point. We would expect that the separatrix running from the Gaussian f.p. to the third f.p. in (29) (which is similar to tri-critical f.p. in the language of statistical physics) should determine the cross-over to a region with fluctuation-induced first order phase transition. More specifically the RG flow in this region should end up in an ordered phase. In the next section we will argue that this is indeed the case.

## 3.2 Coleman-Weinberg mechanism

In this section we will follow the standard reasoning of Coleman and Weinberg [21] applied within the context of this paper. We will expand around classical field values

$$
\begin{aligned}
\phi^i &= \phi_{cl}^i + \delta\phi^i, \quad \left|\delta\phi^i\right| \ll 1, \\
\chi^a &= \chi_{cl}^a + \delta\chi^a, \quad |\delta\chi^a| \ll 1,
\end{aligned}
\tag{30}
$$

and only keep up to quadratic terms

$$
S = S[\phi_{cl}, \chi_{cl}] + \int_{\mathbb{R}_+^d} d^d x \left( \frac{(\partial\delta\phi)^2}{2} + \frac{(\partial\delta\chi)^2}{2} + \delta(x_\perp)\delta V(\phi_{cl}^2, \delta\phi^2, \chi_{cl}^2, \delta\chi^2) \right),
\tag{31}
$$

where the quadratic part of the potential can be written as a boundary mass term

$$
\delta V = \Phi^I m_\Phi^{IJ} \Phi^J,
\tag{32}
$$

with

$$
m_\Phi^{IJ} = \begin{pmatrix} m_\phi^{ij} & g\phi_{cl}^j \chi_{cl}^b \\ g\chi_{cl}^a \phi_{cl}^k & m_\chi^{ab} \end{pmatrix}^{IJ},
\tag{33}
$$

$$
\begin{aligned}
m_\phi^{ij} &\equiv A_g^\lambda \delta^{ij} + \lambda \hat{\phi}_{cl}^i \hat{\phi}_{cl}^j, \\
m_\chi^{ab} &\equiv A_\lambda^g \delta^{ab} + \lambda \hat{\chi}_{cl}^a \hat{\chi}_{cl}^b, \\
A_y^x &= \frac{x\phi_{cl}^2 + y\chi_{cl}^2}{2}.
\end{aligned}
\tag{34}
$$

Here we defined the field $\Phi^I = \delta\phi^j \oplus \delta\chi^a$, $I, J \in \{1, \ldots, 2N\}$. The one-loop correction to the path integral $Z_\Phi$ can be calculated by substituting the above mass term into the formula derived in section 2 and performing the relevant momentum integral (23). We leave the details of this computation to appendix C. It yields the effective boundary potential

$$
\begin{aligned}
V_{\text{eff}}(\phi_{cl}^2, \chi_{cl}^2) = &\left( \frac{\lambda}{8} + B_1 \right)\phi_{cl}^4 + \left( \frac{g}{4} + B_2 \right)\phi_{cl}^2 \chi_{cl}^2 + \left( \frac{\lambda}{8} + B_1 \right)\chi_{cl}^4 \\
&+ \Xi_1(\phi_{cl}^2, \chi_{cl}^2) + \Xi_2(\phi_{cl}^2, \chi_{cl}^2) + A_1\phi_{cl}^2 + A_2\chi_{cl}^2,
\end{aligned}
\tag{35}
$$

where $\Xi_1$, $\Xi_2$ can be found in appendix C, and the constants $A_i$, $B_i$, $i \in \{1,2\}$ are counter-terms (which depend on the momentum cut-off $\Lambda$) which can be fixed by defining the renormalised masses and coupling constants as the respective coefficients in the potential

$$
\frac{\partial V}{\partial(\phi_{cl}^2)}\bigg|_{\phi_{cl}^2=\chi_{cl}^2=0} = \frac{\partial V}{\partial(\chi_{cl}^2)}\bigg|_{\phi_{cl}^2=\chi_{cl}^2=0} = 0,
$$

$$
\frac{\partial^2 V}{\partial(\phi_{cl}^2)^2}\bigg|_{\phi_{cl}^2=\chi_{cl}^2=0} = \frac{\partial^2 V}{\partial(\chi_{cl}^2)^2}\bigg|_{\phi_{cl}^2=\chi_{cl}^2=0} = \frac{\lambda}{4},
$$
$$
\frac{\partial^2 V}{\partial(\phi_{cl}^2)\partial(\chi_{cl}^2)}\bigg|_{\phi_{cl}^2=\chi_{cl}^2=0} = \frac{g}{4}.
\tag{36}
$$

The latter two derivatives are IR divergent in the $\phi_{cl}$, $\chi_{cl} \to 0$ limit due to the presence of logarithms in $V_{\text{eff}}$. Following the CW procedure we can resolve this issue by evaluating the renormalisation conditions at non-zero field value for $\phi$ (alternately for $\chi$)

$$
\frac{\partial V_{\text{eff}}}{\partial(\phi_{cl}^2)}\bigg|_{\phi_{cl}^2=\chi_{cl}^2=0} = \frac{\partial V_{\text{eff}}}{\partial(\chi_{cl}^2)}\bigg|_{\phi_{cl}^2=\chi_{cl}^2=0} = 0,
$$

$$
\frac{\partial^2 V_{\text{eff}}}{\partial(\phi_{cl}^2)^2}\bigg|_{\phi_{cl}^2=\mu,\chi_{cl}^2=0} = \frac{\partial^2 V_{\text{eff}}}{\partial(\chi_{cl}^2)^2}\bigg|_{\phi_{cl}^2=\mu,\chi_{cl}^2=0} = \frac{\lambda}{4},
$$
$$
\frac{\partial^2 V_{\text{eff}}}{\partial(\phi_{cl}^2)\partial(\chi_{cl}^2)}\bigg|_{\phi_{cl}^2=\mu,\chi_{cl}^2=0} = \frac{g}{4},
\tag{37}
$$

where $\mu$ is an arbitrary RG scale and we used that near $d=3$ the scaling dimension of $\phi_c$ is (27) so to leading order in $\epsilon$-expansion $\phi_{cl}^2$ scales as mass.[16]

The renormalisation conditions (37) now fix the counter-terms in such a way that the divergences in $\Lambda$ vanish in the effective potential

$$
A_1 = (d-4)e^{(d-3)\gamma_E/2}\frac{Ng + (N+2)\lambda}{2^d \pi^{(d-1)/2}}\Lambda^{d-2},
$$
$$
A_2 = (d-4)e^{(d-3)\gamma_E/2}\frac{Ng + (N+2)\xi}{2^d \pi^{(d-1)/2}}\Lambda^{d-2},
\tag{38}
$$

$$
B_1 = \left(\frac{N+8}{4\pi}\lambda^2 + \frac{N}{4\pi}g^2\right)\frac{\log\Lambda}{8} + \dots,
$$
$$
B_2 = \left(\frac{g^2}{\pi} + 2\frac{N+2}{4\pi}\lambda g\right)\frac{\log\Lambda}{4} + \dots
\tag{39}
$$

Here we only wrote out the divergent parts of the bare couplings (75) in $B_i$. As a consistency check we can readily verify that the coefficients of the logarithmic divergences in $B_i$ agree with the $\beta$-functions (28) computed with dimensional regularisation. If we plug these constants into (35) we get the full effective potential which we do not write out here since it is given by a cumbersome expression. Details of this can be found in an appended Mathematica notebook.

---

[16]Note that we choose to define the renormalisation conditions w.r.t. $\phi_{cl}^2$, $\chi_{cl}^2$ as opposed to some particular component of $\phi_{cl}$, $\chi_{cl}$. In this way we obtain $O(N)$-invariant counter-terms, but otherwise the physics remains the same.

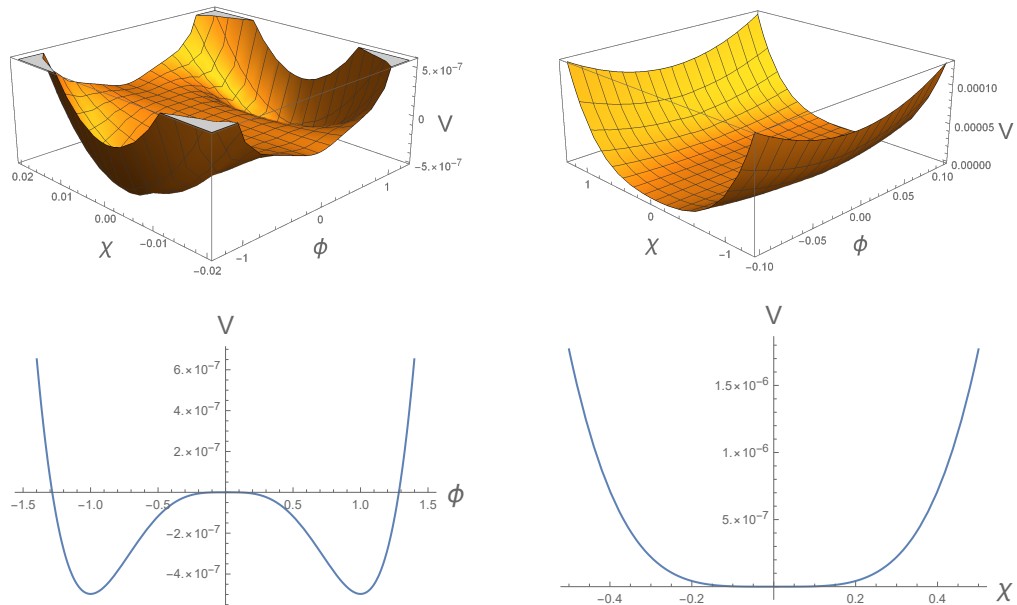

Figure 3: Plots of the effective potential for $N = 1$. There are two three-dimensional plots: one with narrow range of $\phi_{cl}$ and one of $\chi_{cl}$. We can see that the potential only has two minima along the $\chi_{cl}$-axis. The two-dimensional plots are slices of the three-dimensional plot when $\phi_{cl} = 0$ or $\chi_{cl} = 0$. In the plots $g = 0.01$ and $\mu = 1$.

We can verify by explicit computation that this effective potential admits a perturbative minimum at $\phi_{cl}^2 = \mu$ with

$$\phi_{cl}^i \left. \frac{\partial V}{\partial \hat{\phi}_{cl}^i} \right|_{\phi_{cl}^2 = \langle \phi \rangle^2 = \mu, \chi_{cl}^2 = \langle \chi \rangle^2 = 0} = \chi_{cl}^a \left. \frac{\partial V}{\partial \chi_{cl}^a} \right|_{\phi_{cl}^2 = \langle \phi \rangle^2 = \mu, \chi_{cl}^2 = \langle \chi \rangle^2 = 0} = 0, \qquad (40)$$

provided the couplings satisfy the relation

$$\lambda = \frac{4\pi - \sqrt{16\pi^2 - 4N(N+8)g^2}}{2(N+8)} = \frac{Ng^2}{4\pi} + \mathcal{O}(g^3). \qquad (41)$$

This relation describes a region very close to the Gaussian f.p. (rather than a WF one), making it independent of the $\epsilon$-expansion. A plot of the effective potential with $N = 1$ is depicted on figure 3 from which we can see that this solution indeed corresponds to a minimum. Without loss of generality we can parametrise this solution as follows

$$\langle \phi \rangle = (\sqrt{\mu}, 0, \dots 0), \quad \langle \chi \rangle = 0. \qquad (42)$$

This minimum tells us that the $O(N) \oplus O(N)$-symmetry has been broken into $O(N-1) \oplus O(N)$. Additionally this vacuum breaks the discrete symmetry (26).

Since this vacuum only breaks one of the $O(N)$-symmetries we can apply the arguments discussed in the introduction around (4). In particular we can now study the perturbations around (42) by using the parametrisation (6) for $\phi$. Expanding the effective potential to the quadratic order yields[17]

$$V_{\text{eff}}(\sigma, \chi^2) = \frac{Ng^2\mu}{8\pi}\sigma^2 + \left(1 - \frac{g}{\pi}\right)\frac{g\mu}{4}\chi^2 + \dots, \qquad (43)$$

---

[17]At higher orders there will be interactions with both even and odd powers of $\sigma$, e.g. $\sigma^3$ and $\sigma\chi_{cl}^2$.

where the dots stand for higher order terms in $g$, $\chi$, $\sigma$. The positive sign of both mass terms is a consequence of (42) being the minimum of the effective potential. The leading (positive) correction to the mass term for $\chi$ is a purely classical consequence of the mixed coupling. Hence we see that the potential (43) induces a boundary RG flow ending with $N-1$ Neumann scalars from the broken $O(N)$-symmetry.

To summarize, the theory (24) we started with had $O(N) \oplus O(N)$-symmetry as well as the symmetry (26). After integrating out quantum fluctuations, one of the $O(N)$-symmetries is still preserved while the symmetry (26) is completely broken and the other $O(N)$-symmetry is broken down to its subgroup $O(N-1)$. The remaining $O(N-1)$ can be seen through the effective theory in the IR which contains $N-1$ Neumann scalars (which additionally regain the shift symmetry (3)), and $N+1$ Dirichlet scalars.

At last let us discuss the validity of the one-loop approximation and its relevance to the flow diagram charted on figure 2. The condition (41) tells us that the region of validity of the approximation lies in the $\lambda$, $g > 0$ quadrant. Furthermore, in the $g \ll 1$ limit this region lies below the line connecting the Gaussian f.p. with the 3rd TC f.p. in (29), which is defined by the relation $\lambda = kg$ with $k$ being $\mathcal{O}(g^0)$ and positive. As we can see in figure 2, the flow in this region drives the coupling $\lambda$ to negative values and hence we would indeed expect a phase-transition happening here. We should also remark that the approximation we used cannot be trusted for field values far from $\sqrt{\mu}$ and thus we cannot exclude the possibility of other vacua hiding in these regions.

One might also wonder whether the phase transition persists in the $\epsilon \to 1$ limit given the applicability of Mermin-Wagner theorem mentioned in the introduction. While we presently cannot give a definite answer to this question we can make the following qualitative observation: from the discussion around (29) we see that the phase diagram in figure 2 exists only for models with low $N$ $(< 4)$. This window certainly includes $N = 1$ and one might expect that it shrinks further at higher orders in $\epsilon$. Thus it could be that in the $\epsilon \to 1$ limit only the $N = 1$ remains in which case the broken symmetry in question is discrete ($\mathbb{Z}_2$) which is not at odds with the Mermin-Wagner theorem.

Let us finally mention the $d = 3$ case. For $\epsilon = 0$ the three non-Gaussian f.p.'s in figure 2 disappear and the asymptotic freedom is lost.[18] Despite that, the arguments of this subsection apply if we think of the model at non-zero $(g, \lambda)$ as an effective field theory with radiately generated potential just as in the original Coleman-Weinberg paper.

## 4   Conclusion

In this paper we have argued that many of the critical phenomena appearing for interacting bulk systems can also be observed in free theories with non-trivial dynamical b.c.'s. These dynamical b.c.'s generically break the conformal symmetry and induce an RG flow at the boundary. We have found that in this context the phase transitions should be understood in terms of the b.c.'s at the IR end of this flow. The second-order phase transitions are described by a boundary RG flow preserving the global symmetries of the theory. It has an IR f.p. with conformal b.c.'s that are neither Dirichlet nor Neumann. To check whether the f.p.'s we discovered in section 3 are artefacts of the $\epsilon$-expansion or actual physical boundary CFT's would require a non-perturbative approach which is beyond the scope of this paper. An evidence for existence of such f.p. beyond perturbation theory was nevertheless put forward in a recent work [16] employing the numerical boostrap. It would be interesting to investigate the existence of the phase diagram 2 by such boostrap methods.

Our analysis also suggests the possibility of RG flows leading to first-order phase transi-

---

[18]More concretely the boundary RG flow ends with the Gaussian f.p. with Neumann b.c.'s for all fields.

tions induced by quantum effects. These will be described by a combination of Dirichlet and Neumann scalars, with the latter playing a role analogous to Goldstone bosons of the ordinary symmetry breaking. To confirm such assertion beyond the perturbative reasoning offered here, we could devise a lattice simulation of the model.

The physical interpretation of the model described in section 3 remains an open question. It would be very interesting to explore whether the interpolation $\epsilon \to 1$ of the model we described in section 3 describes a meaningful two-dimensional theory. In particular there remains the question of whether the fixed points (29) correspond to conformal boundary conditions in $d = 2$. For compact scalars in $d = 2$ there is a body of evidence [28, 29] suggesting that in a conformal theory we can have at most a combination of Dirichlet and Neumann boundary conditions. To make a connection with our work we note that in $d = 2$ the full potential in (24) should include infinitely many more terms that are classically marginal. Such deformations do lead to f.p.'s corresponding to Dirichlet b.c.'s [13, 30] and hence it would not be unreasonable to expect something similar to happen here too.

In $d = 3$ the free scalar can be interpreted as a dual photon of the Maxwell theory. A boundary potential (1) would correspond to a monopole potential breaking the topological $U(1)$ symmetry. Given that the bulk theory is free, it would be very interesting to investigate the possibility of exactly solvable monopole potentials.

The free $O(N)$ model with $N > 1$ also has a nice condensed matter interpretation as crystaline displacement fields with $N$ being the spatial dimension of the solid [31]. The boundary potential we consider would correspond to dislocations interacting directly at the edge of the solid. It would be amusing to explore whether it can describe a realistic physical situation.

Let us mention a few interesting possible extensions of this work. First we could try coupling the free bulk scalar to boundary degrees of freedom and use this to generate an effective potential and condensates for the boundary fields. This could provide some quantitative arguments for the possible existence of ordered phases of mixed dimensional theories similar to the ones recently considered in the literature (e.g. [32–34]).

On the other hand we could consider adding bulk couplings and making connection with the recent work [8], where the contribution of a bulk $\phi^6$-interaction to the one-loop effective action was computed.

# Acknowledgements

We are grateful to Hans Werner Diehl, Guido Festuccia and Chris Herzog for illuminating discussions and comments as well as to Agnese Bissi for reading the manuscript. VP was supported by the ERC STG grant 639220 during the work on this project and Vetenskapsrådet under grant 2018-05572 in the later stages. AS is supported by Knut and Alice Wallenberg Foundation KAW 2016.0129.

# A  Propagators in boundary quantum field theories

In this appendix we will study Feynman propagators in an $O(N)$-symmetric scalar BQFT with masses in both the bulk and on the boundary

$$\mathcal{L} = \int_{\mathbb{R}^d_+} d^d x \left( \frac{(\partial_\mu \phi^i)^2}{2} + \frac{(m^2)^{ij} \phi^i \phi^j}{2} + \delta(x_\perp) \hat{m}^{ij} \phi^i \phi^j \right). \tag{44}$$

This theory has the e.o.m.

$$\left(-\delta^{ij}\partial^2 + (m^2)^{ij}\right)\phi^j = 0,$$
$$\partial_\perp \phi^i|_{x_\perp=0} = M^{ij}\phi^j|_{x_\perp=0}. \tag{45}$$

Please note that the RG flow of the boundary mass describes a flow between Neumann ($M^{ij} = 0$) and Dirichlet ($M^{ij} \to \infty$) b.c.'s.

## A.1 In spacetime coordinates

We will find the Feynman propagator $\Delta^{ij}_{bc}(s_\parallel, x_\perp, y_\perp) \equiv \langle \phi^i(x_\parallel, x_\perp)\phi^j(y_\parallel, y_\perp)\rangle$. It satisfies the Dyson-Schwinger equation corresponding to the e.o.m. (45) at separate points

$$\left(-\delta^{ij}\partial_x^2 + (m^2)^{ij}\right)\Delta^{jk}_{bc}(s_\parallel, x_\perp, y_\perp) = \delta^{ik}\delta(s_\parallel, x_\perp - y_\perp),$$
$$\partial_{x_\perp}\Delta^{ik}_{bc}(s_\parallel, 0, y_\perp) = M^{ij}\Delta^{jk}_{bc}(s_\parallel, 0, y_\perp). \tag{46}$$

Here $\partial_x^2 \equiv \partial_{x_\parallel}^2 + \partial_{x_\perp}^2$ is the d'Alembert operator and $s_\parallel \equiv x_\parallel - y_\parallel$ is the distance between the parallel coordinates. Let us first study the first equation in a homogeneous QFT

$$\left(-\delta^{ij}\partial_x^2 + (m^2)^{ij}\right)\Delta^{jk}(s) = \delta^{ik}\delta(s), \tag{47}$$

where $s \equiv s_\parallel^a \oplus (x_\perp - y_\perp) \in \mathbb{R}^d$. To solve this consider the Fourier-transform of $\Delta^{jk}(s)$

$$\Delta^{jk}(s) = \int_{\mathbb{R}^d} \frac{d^d k}{(2\pi)^d} e^{-iks} G^{jk}(k), \tag{48}$$

which yields

$$\partial^2 \Delta^{jk}(s) = \int_{\mathbb{R}^d} \frac{d^d k}{(2\pi)^d} e^{-iks}(-ik)^2 G^{jk}(k), \tag{49}$$

and use the definition of the Dirac $\delta$-function

$$\delta(s) = \int_{\mathbb{R}^d} \frac{d^d k}{(2\pi)^d} e^{-iks}. \tag{50}$$

We can now compare the integrands in (47) to find

$$G^{jk}(k) = \left(\delta^{jk}k^2 + (m^2)^{jk}\right)^{-1}. \tag{51}$$

Finding this inverse is difficult without knowing the specific form of $m^{ij}$. Let us assume it is on the form

$$(m^2)^{ij} = m_1^2 \delta^{ij} + a\phi_{cl}^i\phi_{cl}^j, \tag{52}$$

where $a$, $m_1^2 \in \mathbb{R}$ are two constants and $\phi_{cl}^i$ is a classical background. The momentum propagator (51) is then[19]

$$G^{jk}(k) = \frac{\delta^{jk}}{k^2 + m_1^2} - \frac{a\phi_{cl}^i\phi_{cl}^j}{(k^2+m_1^2)(k^2+m_2^2)}, \quad m_2^2 \equiv m_1^2 + a\phi_{cl}^2. \tag{53}$$

---

[19]Here we used the ansatz $\left(\delta^{ij}k^2 + m^{ij}\right)^{-1} = a\delta^{ij} + b\phi^i\phi^j$, and then found the coefficients $a$ and $b$ by matching $\delta^{ij}$ and $\phi^i\phi^j$ terms on both sides of $\left(\delta^{ij}k^2 + m^{ij}\right)^{-1}\left(\delta^{jk}k^2 + m^{jk}\right) = \delta^{ik}$.

Here $\phi_{cl}^2 \equiv (\phi_{cl}^i)^2$. In the massless limit the second term vanishes. Using Schwinger and Feynman parametrisations, we find $\Delta^{jk}(s)$ by performing the integrals in (48)

$$
\begin{aligned}
I_\alpha^d(m^2) &\equiv \int_{\mathbb{R}^d} \frac{d^d k}{(2\pi)^d} \frac{e^{-iks}}{(k^2+m^2)^\alpha} = \int_0^\infty \frac{du}{\Gamma_\alpha} \int_{\mathbb{R}^d} \frac{d^d k}{(2\pi)^d} u^{\alpha-1} e^{-u(k^2+m^2)-iks} \\
&= \int_0^\infty \frac{du}{(4\pi)^{d/2}\Gamma_\alpha} \frac{e^{-m^2 u - s^2/(4u)}}{u^{(d+2)/2-\alpha}} = \frac{m^{d/2-\alpha} K_{d/2-\alpha}(m|s|)}{2^{(d-2)/2+\alpha}\pi^{d/2}\Gamma_\alpha |s|^{d/2-\alpha}},
\end{aligned}
\tag{54}
$$

$$
\begin{aligned}
J^d(m_1^2, m_2^2) &\equiv \int_{\mathbb{R}^d} \frac{d^d k}{(2\pi)^d} \frac{e^{-iks}}{(k^2+m_1^2)(k^2+m_2^2)} = \int_0^1 du\, I_2^d(u m_1^2 + (1-u)m_2^2) \\
&= -\frac{m_1^{\Delta_\phi} K_{\Delta_\phi}(m_1|s|) - m_2^{\Delta_\phi} K_{\Delta_\phi}(m_2|s|)}{(2\pi)^{d/2} |s|^{\Delta_\phi}(m_1+m_2)(m_1-m_2)}.
\end{aligned}
\tag{55}
$$

This holds for (27). We find

$$
\begin{aligned}
\Delta^{jk}(s) &= \delta^{jk} I_1^d(m_1^2) - a\phi_{cl}^j \phi_{cl}^k J^d(m_1^2, m_2^2) \\
&= \frac{\delta^{jk} m_1^{\Delta_\phi} K_{\Delta_\phi}(m_1|s|)}{(2\pi)^{d/2} |s|^{\Delta_\phi}} + \frac{a\phi_{cl}^j \phi_{cl}^k \left[ m_1^{\Delta_\phi} K_{\Delta_\phi}(m_1|s|) - m_2^{\Delta_\phi} K_{\Delta_\phi}(m_2|s|) \right]}{(2\pi)^{d/2} |s|^{\Delta_\phi}(m_1+m_2)(m_1-m_2)}.
\end{aligned}
\tag{56}
$$

Here $K_\nu(z)$ is a modified Bessel function of the second kind. Please note that the two terms are on the form $|s|^{-\Delta_\phi} K_{\Delta_\phi}(m_j|s|)$, $j \in \{1, 2\}$.

In the massless limit $(m_1, m_2 \to 0)$ this reduces to the usual correlator in a CFT

$$
\lim_{m \to 0} \Delta^{jk}(s) = \frac{A_d \delta^{jk}}{|s|^{2\Delta_\phi}}, \quad A_d = \frac{1}{(d-2)S_d},
\tag{57}
$$

where $S_d$ is the area of a $(d-1)$-dimensional sphere.

Let us now proceed with finding the Feynman propagator in a BQFT, i.e. we wish to use (56) to solve the b.c. in (46). To do this we make the ansatz[20]

$$
\Delta_{bc}^{ik}(s_\parallel, x_\perp, y_\perp) = \Delta^{ik}(s_\parallel, x_\perp - y_\perp) + \Delta^{ik}(s_\parallel, x_\perp + y_\perp) + \int_0^\infty dz\, f^{ij}(z) \Delta^{jk}(s_\parallel, x_\perp + y_\perp + z).
$$

We want to find the function $f^{ij}(z)$ from the b.c. (46). Since all terms in (56) are on the same form, the function $f^{ij}(z)$ will be the same for all of them. This means we can let $\Delta^{ij}(s_\parallel, x_\perp) = \tau^{ij}(s_\parallel^2 + x_\perp^2)^{-\Delta_\phi/2} K_{\Delta_\phi}(m\sqrt{s_\parallel^2 + x_\perp^2})$, with the tensor structure $\tau^{ij} \in \{\delta^{ij}, \phi_{cl}^i \phi_{cl}^j\}$, when we find $f^{ij}(z)$. The second term in the ansatz above reduces the normal derivative in the LHS of the b.c. to a single integral using the recursion relation

$$
K_{\Delta_\phi - 1}(z) = K_{\Delta_\phi + 1}(z) - \frac{2\Delta_\phi}{z} K_{\Delta_\phi}(z).
\tag{58}
$$

We find

$$
\begin{aligned}
&\partial_{x_\perp} \Delta_{bc}^{ik}(s_\parallel, 0, y_\perp) \\
&= -m \int_0^\infty dz\, f^{ij}(z) \tau^{jk} \frac{z_\perp + y_\perp}{\left(s_\parallel^2 + (z_\perp + y_\perp)^2\right)^{(\Delta_\phi+1)/2}} K_{\Delta_\phi+1}(m\sqrt{s_\parallel^2 + (z_\perp + y_\perp)^2}) \\
&= -f^{ij}(0) \Delta^{jk}(s_\parallel, y_\perp) - \int_0^\infty dz\, \partial_z f^{ij}(z) \Delta^{jk}(s_\parallel, z_\perp + y_\perp).
\end{aligned}
\tag{59}
$$

---

[20] Here we are using the method of images where we consider a semi-infinite line of images on the other side of the boundary. Each image corresponds to adding a Dirac $\delta$-function that is always zero (since $x_\perp, y_\perp > 0$) on the RHS of (46).

Here we used partial integration, and assumed that

$$\lim_{z \to \infty} \frac{e^{-mz} f^{ij}(z)}{z^{\Delta_\phi + 1/2}} = 0. \tag{60}$$

The RHS of the b.c. at (46) is

$$M^{ij} \Delta^{jk}_{bc}(s_\parallel, 0, y_\perp) = 2M^{ij} \Delta^{jk}(s_\parallel, y_\perp) + \int_0^\infty dz M^{ij} f^{jl}(z) \Delta^{lk}(s_\parallel, z_\perp + y_\perp). \tag{61}$$

We can now match the terms outside and inside the integral with those in (59) to find an ordinary differential equation for $f^{ij}(z)$. It tells us that $f^{ij}(z)$ is an exponential of the matrix $M^{ij}$

$$\left\{ \begin{array}{ll} f^{ij}(0) & = -2M^{ij} \\ \partial_z f^{ij}(z) & = -M^{ik} f^{kj}(z) \end{array} \right\} \Rightarrow \quad f^{ij}(z) = -2M^{ik}(e^{-Mz})^{kj}. \tag{62}$$

This function does indeed satisfy (60). It gives us the Feynman propagator

$$\boxed{\begin{aligned} \Delta^{ij}_{bc}(s_\parallel, x_\perp, y_\perp) = {}& \Delta^{ij}(s_\parallel, x_\perp - y_\perp) + \Delta^{ij}(s_\parallel, x_\perp + y_\perp) \\ & - 2M^{ik} \int_0^\infty dz (e^{-Mz})^{kl} \Delta^{lj}(s_\parallel, x_\perp + y_\perp + z). \end{aligned}} \tag{63}$$

## A.2 Momentum propagator

In this appendix we will Fourier transform the Feynman propagator found in the previous section with respect to $s_\parallel$. We call this a momentum propagator, although it depends on the normal coordinates $x_\perp$ and $y_\perp$, which should be seen as new scales in a BQFT that regulate divergences that may appear in the boundary limit [10]. Since each term (63) behave in the same way w.r.t. $s_\parallel$, we will first study the Fourier transform of

$$\Delta^{jk}(s_\parallel, x_\perp) = \frac{m^{\Delta_\phi} \tau^{jk} K_{\Delta_\phi}(m\sqrt{s_\parallel^2 + x_\perp^2})}{(2\pi)^{d/2}(s_\parallel^2 + x_\perp^2)^{\Delta_\phi/2}}, \quad \tau^{jk} \in \{\delta^{ij}, \phi^j_{cl}\phi^k_{cl}\}. \tag{64}$$

We will use the following representations of the Bessel function

$$\begin{aligned} K_\nu(z) &= \int_{\gamma_n} \frac{dt}{4\pi i} \Gamma_t \Gamma_{t-\nu} \left(\frac{z}{2}\right)^{\nu - 2t}, \quad \gamma_n = \{t \in \mathbb{R} \mid \nu + \epsilon + it, \, \epsilon > 0\}, \\ K_\nu(z) &= \left(\frac{2}{x}\right)^\nu \frac{\Gamma_{\nu + 1/2}}{\sqrt{\pi}} \int_0^\infty du \frac{\cos(uz)}{(u^2 + 1)^{\nu + 1/2}}. \end{aligned} \tag{65}$$

Here $\Gamma_x \equiv \Gamma(x)$ is the Gamma function. The first of the identities above, together with a Schwinger parametrisation yields

$$\begin{aligned} G^{jk}_\parallel(k_\parallel, x_\perp) &= \int_{\mathbb{R}^{d-1}} d^{d-1} s_\parallel e^{ik_\parallel s_\parallel} \Delta^{jk}(s_\parallel, x_\perp) \\ &= \frac{\tau^{jk}}{\pi^{(d-2)/2} i} \int_{\gamma_{\Delta_\phi}} dt \frac{m^{2(\Delta_\phi - t)} \Gamma_t \Gamma_{t - \Delta_\phi}}{2^{(d+4)/2 + \Delta_\phi - 2t}} \int_{\mathbb{R}^{d-1}} d^{d-1} s_\parallel \frac{e^{ik_\parallel s_\parallel}}{(s_\parallel^2 + x_\perp^2)^t} \end{aligned}$$

$$= \frac{\tau^{jk}}{\pi^{(d-2)/2}i} \int_{\gamma_{\Delta_\phi}} dt \frac{m^{2(\Delta_\phi-t)}\Gamma_{t-\Delta_\phi}}{2^{(d+4)/2+\Delta_\phi-2t}} \int_0^\infty du\, u^{t-1} e^{-ux_\perp^2} \int_{\mathbb{R}^{d-1}} d^{d-1}s_\parallel e^{-us_\parallel^2+ik_\parallel s_\parallel}$$

$$= \frac{\tau^{jk}}{\pi^{3/2}i} \int_{\gamma_{\Delta_\phi}} dt \frac{m^{2(\Delta_\phi-t)}\Gamma_{t-\Delta_\phi}}{2^{(d+4)/2+\Delta_\phi-2t}} \int_0^\infty du \frac{e^{-ux_\perp^2-k_\parallel^2/(4u)}}{u^{(d+1)/2-t}} \tag{66}$$

$$= \frac{\tau^{jk}}{\pi^{3/2}i} \int_{\gamma_{\Delta_\phi}} dt \frac{m^{2(\Delta_\phi-t)}\Gamma_{t-\Delta_\phi}}{2^{\Delta_\phi-t+3/2}} \left(\frac{x_\perp}{|k_\parallel|}\right)^{(d-1)/2-t} K_{(d-1)/2-t}(|k_\parallel|x_\perp).$$

This integrand has simple poles at $t = \Delta_\phi - n$, $n \in \mathbb{Z}_{\geq 0}$ coming from $\Gamma_{t-\Delta_\phi}$. As dictated by the residue theorem, we need to sum over all of the corresponding residues. In order to do this summation, we use the second identity in (65)

$$G_\parallel^{jk}(k_\parallel, x_\perp) = \frac{\tau^{jk}}{\sqrt{\pi}} \sum_{n\geq 0} \frac{(-1)^n m^{2n}}{2^{n+1/2}n!} \left(\frac{x_\perp}{|k_\parallel|}\right)^{(d-1)/2-\Delta_\phi+n} K_{(d-1)/2-\Delta_\phi+n}(|k_\parallel|x_\perp)$$

$$= \frac{2^{(d-2)/2-\Delta_\phi}\tau^{jk}}{\pi} \int_0^\infty du \cos(u|k_\parallel|x_\perp) \sum_{n\geq 0} \frac{(-1)^n m^{2n}\Gamma_{d/2-\Delta_\phi+n}}{n!|k_\parallel|^{d-1-2\Delta_\phi+2n}(u^2+1)^{d/2-\Delta_\phi+n}}$$

$$= \frac{2^{(d-2)/2-\Delta_\phi}\Gamma_{d/2-\Delta_\phi}|k_\parallel|\tau^{jk}}{\pi} \int_0^\infty du \frac{\cos(u|k_\parallel|x_\perp)}{(k_\parallel^2 u^2+k_\parallel^2+m^2)^{d/2-\Delta_\phi}}$$

$$= \frac{\tau^{jk} x_\perp^{(d-1)/2-\Delta_\phi} K_{\Delta_\phi-(d-1)/2}(\sqrt{k_\parallel^2+m^2}x_\perp)}{\sqrt{2\pi}(k_\parallel^2+m^2)^{(d-1)/4-\Delta_\phi/2}}. \tag{67}$$

This simplifies drastically in the case of fundamental scalars (27)

$$G_\parallel^{jk}(k_\parallel, x_\perp) = \frac{\tau^{jk} e^{-\sqrt{k_\parallel^2+m^2}x_\perp}}{2\sqrt{k_\parallel^2+m^2}}. \tag{68}$$

Let us now perform the integration over $z$ in (63)

$$\tilde{G}^{ij}(k_\parallel, x_\perp+y_\perp) = -2M^{ik} \int_0^\infty dz \left(e^{-Mz}\right)^{kl} G_\parallel^{lj}(k_\parallel, x_\perp+y_\perp+z)$$

$$= -\frac{M^{ik}}{\sqrt{k_\parallel^2+m^2}} \sum_{n\geq 0} \frac{(M^n)^{kl}\tau^{lj}}{n!} \int_0^\infty dz(-z)^n e^{-\sqrt{k_\parallel^2+m^2}(x_\perp+y_\perp+z)}$$

$$= -e^{-\sqrt{k_\parallel^2+m^2}(x_\perp+y_\perp)} M^{ik} \sum_{n\geq 0} \frac{(-1)^n(M^n)^{kl}}{(k_\parallel^2+m^2)^{n/2+1}}\tau^{lj}$$

$$= -\frac{e^{-\sqrt{k_\parallel^2+m^2}(x_\perp+y_\perp)}}{\sqrt{k_\parallel^2+m^2}} \left(M^{ik}+\sqrt{k_\parallel^2+m^2}\delta^{ik}-\sqrt{k_\parallel^2+m^2}\delta^{ik}\right)\left(M^{kl}+\sqrt{k_\parallel^2+m^2}\delta^{kl}\right)^{-1}\tau^{lk}$$

$$= e^{-\sqrt{k_\parallel^2+m^2}(x_\perp+y_\perp)} \left(-\frac{\tau^{ij}}{\sqrt{k_\parallel^2+m^2}}+\left(M^{ik}+\sqrt{k_\parallel^2+m^2}\delta^{ik}\right)^{-1}\tau^{kj}\right).$$

Together with (68), we now have the full Fourier transform of (63)

$$H_m^{ik}(k_\parallel, x_\perp, y_\perp) \equiv \left[G_\parallel^{ij}(K_\parallel, x_\perp-y_\perp)+G_\parallel^{ij}(K_\parallel, x_\perp+y_\perp)+\tilde{G}^{ij}(k_\parallel, x_\perp+y_\perp)\right](\tau^{jk})^{-1}$$

$$= \frac{e^{-\sqrt{k_\parallel^2+m^2}x_\perp}\sinh(\sqrt{k_\parallel^2+m^2}y_\perp)\delta^{ik}}{\sqrt{k_\parallel^2+m^2}}+e^{-\sqrt{k_\parallel^2+m^2}(x_\perp+y_\perp)}\left(M^{ik}+\sqrt{k_\parallel^2+m^2}\delta^{ik}\right)^{-1}.$$

This means that the Fourier transform of (63), with (56) is given by

$$G_{bc}^{ik}(k_\parallel, x_\perp, y_\perp) = H_{m_1}^{ik}(k_\parallel, x_\perp, y_\perp) + a\phi^i\phi^j \frac{H_{m_1}^{jk}(k_\parallel, x_\perp, y_\perp) - H_{m_2}^{jk}(k_\parallel, x_\perp, y_\perp)}{(m_1 + m_2)(m_1 - m_2)}.$$

In the boundary limit we have

$$H_m^{ik}(k_\parallel, 0, 0) = \left(M^{ik} + \sqrt{k_\parallel^2 + m^2}\,\delta^{ik}\right)^{-1}. \tag{69}$$

# B $\beta$-function

In this appendix we find the $\beta$-functions for the model (24). This is done in the standard QFT way. We will study the following correlators up to order two in the coupling constants

$$
\begin{aligned}
G_\phi^{jklm}(p) &= \langle \hat{\phi}^j(p_1)\hat{\phi}^k(p_2)\hat{\phi}^l(p_3)\hat{\phi}^m(p_4)\rangle \\
&= -\frac{\lambda_0}{8}8D^{jklm} + \left(-\frac{\lambda_0}{8}\right)^2 32\frac{E^{jklm}I_{12} + E^{jlkm}I_{13} + E^{jmkl}I_{14}}{(2\pi)^{d-1}} \\
&\quad + \left(-\frac{g_0}{4}\right)^2 8\frac{\delta^{jk}\delta^{lm}I_{12} + \delta^{jl}\delta^{km}I_{13} + \delta^{jm}\delta^{kl}I_{14}}{(2\pi)^{d-1}} + \dots,
\end{aligned}
\tag{70}
$$

$$
\begin{aligned}
G_\chi^{abcd}(p) &= \langle \hat{\chi}^a(p_1)\hat{\chi}^b(p_2)\hat{\chi}^c(p_3)\hat{\chi}^d(p_4)\rangle \\
&= -\frac{\xi_0}{8}8D^{abcd} + \left(-\frac{\xi_0}{8}\right)^2 32\frac{E^{abcd}I_{12} + E^{acbd}I_{13} + E^{adbc}I_{14}}{(2\pi)^{d-1}} \\
&\quad + \left(-\frac{g_0}{4}\right)^2 8\frac{\delta^{ab}\delta^{cd}I_{12} + \delta^{ac}\delta^{bd}I_{13} + \delta^{ad}\delta^{bc}I_{14}}{(2\pi)^{d-1}} + \dots,
\end{aligned}
\tag{71}
$$

$$
\begin{aligned}
G_{\phi\chi}^{jkab}(p) &= \langle \hat{\phi}^j(p_1)\hat{\phi}^k(p_2)\hat{\chi}^a(p_3)\hat{\chi}^b(p_4)\rangle \\
&= -\frac{g_0}{4}4\delta^{jk}\delta^{ab} + \left(-\frac{g_0}{4}\right)^2 16\delta^{jk}\delta^{ab}\frac{I_{13} + I_{14}}{(2\pi)^{d-1}} \\
&\quad + \left(-\frac{g_0}{4}\right)\left(-\frac{\lambda_0}{8}\right)16(N_\phi + 2)\delta^{jk}\delta^{ab}\frac{I_{12}}{(2\pi)^{d-1}} \\
&\quad + \left(-\frac{g_0}{4}\right)\left(-\frac{\xi_0}{8}\right)16(N_\chi + 2)\delta^{jk}\delta^{ab}\frac{I_{34}}{(2\pi)^{d-1}} + \dots
\end{aligned}
\tag{72}
$$

Here $\lambda_0$, $g_0$ and $\xi_0$ are the bare coupling constants that appear in the action (24). Hatted operators denote their respective boundary fields. We have the Wick theorem factors

$$
\begin{aligned}
D^{jklm} &= \delta^{jk}\delta^{lm} + \delta^{jl}\delta^{km} + \delta^{jm}\delta^{kl}, \\
E^{jklm} &= (N_\phi + 2)\delta^{jk}\delta^{lm} + D^{jklm}.
\end{aligned}
\tag{73}
$$

The master integral $I_{jk}$ is found using the Schwinger parametrisation and is given by an Euler-Beta function

$$
\begin{aligned}
I_{jk} &= I_{1/2,1/2}^{d-1}(p_j + p_k), \\
I_{\alpha\beta}^n(p) &= \int_{\mathbb{R}^n} \frac{d^n k}{|p - k|^\alpha |w|^\beta} = \pi^{n/2}\frac{\Gamma_{\alpha+\beta-n/2}}{\Gamma_\alpha \Gamma_\beta}\frac{B_{n/2-\alpha,n/2-\beta}}{|p|^{2(\alpha+\beta)-n}},
\end{aligned}
\tag{74}
$$

where in $d = 3 - \epsilon$ it has a pole in $\epsilon$

$$I^{d-1}_{1/2,1/2}(p) = \frac{1}{2\pi}\left(\frac{1}{\epsilon} + \log\left(\sqrt{\frac{64\pi}{e^{\gamma_E}}}\right) - \log(p)\right) + \mathcal{O}(\epsilon).$$

The bare coupling constants that renormalise these correlators[21] are given by

$$\lambda_0 = \left(\frac{e^{\gamma_E/2}}{64\pi}\right)^{\epsilon/2}\mu^\epsilon\left(\lambda + \frac{N_\phi + 8}{4\pi}\frac{\lambda^2}{\epsilon} + \frac{N_\chi}{4\pi}\frac{g^2}{\epsilon}\right) + \dots,$$

$$\xi_0 = \left(\frac{e^{\gamma_E/2}}{64\pi}\right)^{\epsilon/2}\mu^\epsilon\left(\xi + \frac{N_\chi + 8}{4\pi}\frac{\xi^2}{\epsilon} + \frac{N_\phi}{4\pi}\frac{g^2}{\epsilon}\right) + \dots, \tag{75}$$

$$g_0 = \left(\frac{e^{\gamma_E/2}}{64\pi}\right)^{\epsilon/2}\mu^\epsilon\left(g + \frac{g^2}{\pi\epsilon} + \frac{N_\phi + 2}{4\pi}\frac{\lambda g}{\epsilon} + \frac{N_\chi + 2}{4\pi}\frac{\xi g}{\epsilon}\right) + \dots$$

Here the dots represent terms that have more than two coupling constants, $\mu$ is the renormalisation scale, and $\lambda$, $g$ as well as $\xi$ are renormalised coupling constants. Please note that we have used multiplicative renormalisation of $g_0$, and both multiplicative and additive renormalisation of $\lambda_0$ as well as $\xi_0$. We can see that $\lambda_0$ and $\xi_0$ are the same up to flavour numbers. To find the $\beta$-functions we will use

$$\frac{\partial \log \sigma_0}{\partial \log \mu} = 0, \quad \frac{\partial \sigma}{\partial \log \mu} = \beta_\sigma, \quad \frac{\partial \log \sigma}{\partial \log \mu} = \frac{\beta_\sigma}{\sigma}, \tag{76}$$

where $\sigma_0 \in \{g_0, \lambda_0, \xi_0\}$ is any bare coupling, and $\sigma \in \{g, \lambda, \xi\}$ is any renormalised coupling. Taking the logarithm of the coupling constants in (75), and only keeping terms that are quadratic in couplings yields

$$\log \lambda_0 = \epsilon \log \mu + \log \lambda + \frac{N_\phi + 8}{4\pi\epsilon}\lambda + \frac{N_\chi}{4\pi\epsilon}\frac{g^2}{\lambda} + \dots,$$

$$\log \xi_0 = \epsilon \log \mu + \log \xi + \frac{N_\chi + 8}{4\pi\epsilon}\xi + \frac{N_\phi}{4\pi\epsilon}\frac{g^2}{\xi} + \dots, \tag{77}$$

$$\log g_0 = \epsilon \log \mu + \log g + \frac{g}{\pi\epsilon} + \frac{N_\phi + 2}{4\pi\epsilon}\lambda + \frac{N_\chi + 2}{4\pi\epsilon}\xi + \dots$$

Now differentiate these equations w.r.t. $\log \mu$ and use the definitions (76)

$$\epsilon + \frac{\beta_\lambda}{\lambda} + \frac{N_\phi + 8}{4\pi\epsilon}\beta_\lambda + \frac{N_\chi}{4\pi\epsilon}\frac{g}{\lambda}\left(2\beta_g - \frac{g}{\lambda}\beta_\lambda\right) = 0,$$

$$\epsilon + \frac{\beta_\xi}{\xi} + \frac{N_\chi + 8}{4\pi\epsilon}\beta_\xi + \frac{N_\phi}{4\pi\epsilon}\frac{g}{\xi}\left(2\beta_g - \frac{g}{\xi}\beta_\xi\right) = 0, \tag{78}$$

$$\epsilon + \frac{\beta_g}{g} + \frac{\beta_g}{\pi\epsilon} + \frac{N_\phi + 2}{4\pi\epsilon}\beta_\lambda + \frac{N_\chi + 2}{4\pi\epsilon}\beta_\xi = 0.$$

The solution to these equations yields the $\beta$-functions

$$\beta_\lambda = -\epsilon\lambda + \frac{N_\phi + 8}{4\pi}\lambda^2 + \frac{N_\chi}{4\pi}g^2 + \dots,$$

$$\beta_\xi = -\epsilon\xi + \frac{N_\chi + 8}{4\pi}\xi^2 + \frac{N_\phi}{4\pi}g^2 + \dots, \tag{79}$$

$$\beta_g = -\epsilon g + \frac{g^2}{\pi} + \frac{N_\phi + 2}{4\pi}\lambda g + \frac{N_\chi + 2}{4\pi}\xi g + \dots$$

---

[21]And which absorbs the factors of $\gamma_E$ and $\log(\pi)$.

## C  Functional determinants

In this appendix we compute the path integral of a bosonic $O(N_\phi) \oplus O(N_\chi)$-vector that is massless in the bulk, but has a tensor mass $m^{IJ}$ at the boundary. We will not assume any specific form of the boundary mass until it is needed. We will write the fluctuation correction to the boundary potential as

$$V \supset \Phi^I m^{IJ} \Phi^J \,, \tag{80}$$

$$m^{IJ} = \begin{pmatrix} m_\phi^{ij} & g\phi_{cl}^j \chi_{cl}^b \\ g\chi_{cl}^a \phi_{cl}^k & m_\chi^{ab} \end{pmatrix}^{IJ} \,,$$
$$m_\phi^{ij} \equiv A_g^\lambda \delta^{ij} + \lambda \hat\phi_{cl}^i \hat\phi_{cl}^j \,, \tag{81}$$
$$m_\chi^{ab} \equiv A_\xi^g \delta^{ab} + \xi \hat\chi_{cl}^a \hat\chi_{cl}^b \,.$$

The constant $A_y^x$ can be found in (34). In this appendix we will not use the exact form of $A_y^x$, although it is important to remember that it is proportional to the coupling constants. Using the results of section 2, i.e. (23), we have

$$V_{\text{eff}}^{\text{1-loop}} = \int_{\mathbb{R}_+^d} d^d x I_{\mathcal{M}} + \int_{\mathbb{R}^{d-1}} d^{d-1} x_\parallel I_{\partial\mathcal{M}} \,,$$
$$I_{\mathcal{M}} = \int_{\mathbb{R}^d} \frac{d^d k}{(2\pi)^d} \text{tr}_{O(N)} \frac{\log[G^{IJ}(k)]}{2} \,, \tag{82}$$
$$I_{\partial\mathcal{M}} = \int_{\mathbb{R}^{d-1}} \frac{d^{d-1} k_\parallel}{(2\pi)^{d-1}} \text{tr}_{O(N)} \frac{\log[G_{b.c.}^{IJ}(k_\parallel)]}{2} \,.$$

Here we trace over the $O(N)$-indices, $G^{IJ}$ is the momentum propagator (53) in the bulk, and $G_{b.c.}^{IJ}$ is the momentum propagator (69) in the boundary limit

$$G^{IJ}(k) = \frac{\delta^{IJ}}{k^2} \,, \quad G_{b.c.}^{IJ}(k_\parallel) = \left( m^{IJ} + |k_\parallel| \delta^{IJ} \right)^{-1} \,. \tag{83}$$

The logarithm of the bulk propagator is

$$\log[G^{IJ}(k)] = -2\delta^{IJ} \log|k| \,. \tag{84}$$

This allows us to find $I_{\mathcal{M}}$ in (82). We will use spherical coordinates and regulate the divergences using a momentum cutoff $\Lambda$. The integral is of the form

$$J_n(\Lambda) \equiv \int_0^\Lambda dr\, r^{n-1} \log(r) = \frac{\Lambda^n}{n} \left( \log(\Lambda) - \frac{1}{n} \right) \,. \tag{85}$$

Which yields

$$I_{\mathcal{M}} = -\frac{(N_\phi + N_\chi) S_d}{(2\pi)^d} J_d(\Lambda) \,. \tag{86}$$

To compute $I_{\partial\mathcal{M}}$ we will use that the logarithm of the inverse of a matrix can be expressed in terms the original matrix via

$$\left\{ \begin{array}{ll} A & = e^{\log(A)} \\ A^{-1} & = e^{\log(A^{-1})} \end{array} \; \Rightarrow \; A^{-1} = e^{-\log(A)} \right\} \Rightarrow \quad \log(A^{-1}) = -\log(A) \,. \tag{87}$$

Using this we find the trace of the logarithm of the momentum propagator (83)

$$\log[G_{b.c.}^{IJ}(k_\parallel)] = -\log\left[|k_\parallel|\delta^{IJ}\left(\frac{m^{IJ}}{|k_\parallel|} + \mathbb{1}\right)\right] = -\delta^{IJ}\log(|k_\parallel|) - \log\left(\frac{m^{IJ}}{|k_\parallel|} + \mathbb{1}\right). \tag{88}$$

To find the second logarithm we diagonalise $m^{IJ}$. It has four eigenvalues. The first two of these are

$$\begin{aligned}
\lambda_1 &= A_g^\lambda, \quad \text{(with multiplicity } N_\phi - 1), \\
\lambda_2 &= A_\xi^g, \quad \text{(with multiplicity } N_\chi - 1),
\end{aligned} \tag{89}$$

and the other two have both multiplicity one

$$\lambda_\pm = \frac{A_g^\lambda + \lambda\phi_{cl}^2 + A_\xi^g + \xi\chi_{cl}^2 \pm \sqrt{(A_\xi^g + \xi\chi_{cl}^2 - A_g^\lambda - \lambda\phi_{cl}^2)^2 + 4g^2\phi_{cl}^2\chi_{cl}^2}}{2}. \tag{90}$$

We proceed with diagonalising the boundary mass $m^{IJ}$ using some matrix $A$ (as we will see, the exact form of $A$ does not matter)

$$m^{IJ} = (A^{-1}DA)^{IJ}, \quad D = \text{diag}(\lambda_3^+, A_g^\lambda, \dots, A_g^\lambda, \lambda_3^-, A_\xi^g, \dots, A_\xi^g). \tag{91}$$

The second logarithm in (88) can now be found from its Taylor expansion

$$\begin{aligned}
\log\left(\frac{m^{IJ}}{|k_\parallel|} + \mathbb{1}\right) &= \sum_{n\geq 1}\frac{(-1)^{n+1}}{n|k_\parallel|^n}((A^{-1}DA)^n)^{IJ} = (A^{-1})^{IK}\sum_{n\geq 1}\frac{(-1)^{n+1}}{n|k_\parallel|^n}((D)^n)^{KL}A^{LJ} \\
&= \left(A^{-1}\text{diag}\left(\log\left(\frac{\lambda_3^+}{|k_\parallel|} + 1\right), \dots\right)A\right)^{IJ}.
\end{aligned} \tag{92}$$

Using cyclicity of the trace, we find

$$\begin{aligned}
\text{tr}\log[G_{b.c.}^{IJ}(k_\parallel)] = &-\log(\lambda_+ + |k_\parallel|) - \log(\lambda_- + |k_\parallel|) \\
&-(N_\phi - 1)\log(A_g^\lambda + |k_\parallel|) - (N_\chi - 1)\log(A_\xi^g + |k_\parallel|).
\end{aligned}$$

The boundary integrals in (82) are then of the form

$$\begin{aligned}
I_{\partial\mathcal{M}} &= -K_{\lambda_3^+}(\Lambda) - K_{\lambda_3^-}(\Lambda) - (N_\phi - 1)K_{A_g^\lambda}(\Lambda) - (N_\chi - 1)K_{A_\xi^g}(\Lambda), \\
K_x(\Lambda) &= \frac{S_{d-1}}{2^d\pi^{d-1}}\int_0^\Lambda dr\,r^{d-2}\log(x + r).
\end{aligned} \tag{93}$$

This integral is a $_2F_1$-hypergeometric function. Its expansion in $\epsilon$ in $3-\epsilon$ dimensions is performed using the HypExp mathematica package [35,36]. We will keep terms up to $\mathcal{O}(\epsilon^2)$ and order two in the coupling constants. After this we expand around large $\Lambda$, and neglect terms that goes as $\Lambda^{-1}$

$$\begin{aligned}
I_{\partial\mathcal{M}} &= -\frac{N_\phi + N_\chi}{2^d\pi^{d-1}}S_{d-1}J_{d-1}(\Lambda) - \tilde{K}_\phi(\Lambda) - \tilde{K}_\chi(\Lambda) + \mathcal{O}(\Lambda^{-1}) + \mathcal{O}(\epsilon^3), \\
K_x(\Lambda) &= -\frac{3d^2 - 22d + 43 - 2(d^2 - 8d + 19)}{16}\Lambda^{d-1}\log(\Lambda) - (d-4)\Lambda^{d-2}x + \frac{x^2}{2}\left(\log\left(\frac{x}{\Lambda}\right) - \frac{1}{2}\right).
\end{aligned}$$

This, together with (86) and (82), yields the full path integral over $\Phi$

$$Z = (2\pi)^{d/2}\exp\left(-\frac{N_\phi + N_\chi}{(2\pi)^d}S_dJ_d - \frac{N_\phi + N_\chi}{2^d\pi^{d-1}}S_{d-1}J_{d-1} - \Xi_1(\phi_{cl}^2, \chi_{cl}^2) - \Xi_2(\phi_{cl}^2, \chi_{cl}^2)\right),$$

$$\Xi_1(\phi_{cl}^2, \chi_{cl}^2) = -(d-4)\Lambda^{d-2}(N_\phi A_g^\lambda + N_\chi A_\xi^g + \lambda\phi_{cl}^2 + \xi\chi_{cl}^2)$$

$$= -(d-4)\Lambda^{d-2}\frac{(N_\chi g + (N_\phi + 2)\lambda)\phi_{cl}^2 + (N_\phi g + (N_\chi + 2)\xi)\chi_{cl}^2}{2},$$

$$\Xi_2(\phi_{cl}^2, \chi_{cl}^2) = \frac{N_\phi - 1}{2}(A_g^\lambda)^2\left(\log\left(\frac{A_g^\lambda}{\Lambda}\right) - \frac{1}{2}\right) + \frac{N_\chi - 1}{2}(A_\xi^g)^2\left(\log\left(\frac{A_\xi^g}{\Lambda}\right) - \frac{1}{2}\right) \quad (94)$$

$$+ \frac{\lambda_+^2}{2}\left(\log\left(\frac{\lambda_+}{\Lambda}\right) - \frac{1}{2}\right) + \frac{\lambda_-^2}{2}\left(\log\left(\frac{\lambda_-}{\Lambda}\right) - \frac{1}{2}\right).$$

Note that the constants $A_y^x$ at (34) depend on $\phi_{cl}^2$ and $\chi_{cl}^2$. By taking $N_\phi = N_\chi$ and $\xi = \lambda$ we obtain the result relevant for section 3.2.

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
