# Peer review of "Spontaneous symmetry breaking in free theories with boundary potentials"

_SciPost Physics, doi:SciPost Phys. 11, 035 (2021)_

## Round 1 · Referee Report · Lorenzo Di Pietro · 2021-6-14

Strengths
1- It is shown how to compute perturbatively boundary effective potentials
2- Boundary symmetry-breaking induced by quantum fluctuations is shown to be possible
3 - The consequences of developing a boundary VEV are illustrated in an interesting example
Weaknesses
1- The main new technical ingredient, the perturbative formula for the boundary effective potential, seems to require only a modest generalization of the familiar concept in QFT without boundary
Report
The paper explains, and illustrates with a nice example, the phenomenon of boundary symmetry-breaking for a theory of free bulk scalars with boundary interactions. To my knowledge the calculation of boundary effective potentials is new, and it can be a useful reference for future developments in the study of field theories on spaces with a boundary. In the example of the O(N)+O(N) model, details of the calculation are provided and I have no reason to doubt the correctness. However there is a point in the derivation of the general formula for the potential that is not clear to me: in the regularization of eq. (22) what is the logic that allows one to subtract the denominator inside the logarithm, even though it is dependent on M? If I can subtract such a non-trivial dependence on M, then what stops me from subtracting the whole divergent integral away? If this is clarified, for the reasons above I deem the paper worth of publication on SciPost.
Requested changes
1-The question mentioned in my report about the regularization of (22) should be addressed, adding some explanation about this point ;
2- In the title of section 3 and in the first line of subsection 3.1 an O seems to be missing for the second O(N) group;
3- In the upper right block of the matrix in eq. (32), the indices should be lower case i,j rather than upper case;
4- In (33) a different notation is used when defining the function A and when the function A is used inside the mass matrices. I request that the same notation is used, preferably the function notation, or that at least the notation with subscripts and superscripts is explicitly explained;
5 - The logic for isolating the terms \Xi_1 and \Xi_2 from the others in equation (34) should be explained. They are not UV finite, they are not field-independent, so it is not clear what makes them special compared to the rest.

---

## Round 1 · Referee Report · Anonymous · 2021-6-15

Report
The paper discusses boundary conditions for the free $O(N)$ model in $3-\epsilon$ dimensions. This choice is dictated by the presence of weakly relevant boundary operators when $\epsilon$ is positive and small, which drive short flows away from the non-interacting fixed point and can be studied perturbatively. The two physical situations are $\epsilon=0$ and $\epsilon=1$. In the former case, the perturbations considered in the paper are irrelevant, and the long distance physics is trivial. The latter case is strongly coupled in the present description, but in the spirit of the epsilon expansion one might find information on the phases of the theory by setting $\epsilon=1$ in the perturbative expressions.
The main result of the paper is a proposal for the phase diagram of a theory where a bulk $O(2N)$ global symmetry is explicitly broken to $O(N)\oplus O(N)$, for generic values of the boundary couplings.
This work is in line with a recent research effort aimed at exploring the space of boundary conditions for conformal field theory in various dimensions. Since boundary conditions, even for free theories, are generically not understood, the setup is interesting and the perturbative approach adopted here is a legitimate line of attack. However, there are two main observations the referee would like the authors to address, and a few minor points.
1. Since the non-trivial physics happens at $\epsilon=1$, it is important to understand if, even in principle, the phase diagram proposed in the paper can be qualitatively correct in two dimensions. The diagram involves a phase where the $O(N)\oplus O(N)$ symmetry is enhanced in the IR to $O(2N)$, and a phase where on the contrary it is spontaneously broken to $O(N)\oplus O(N-1)$. One obvious obstruction to such a scenario is the Mermin-Wagner theorem. For $\epsilon>0$, the authors claim that a continuous symmetry can be spontaneously broken on the boundary, which has dimension smaller than two. This is not prohibited, since the theory on the boundary is non local, and there are examples of this fact in the literature, but the authors should comment on this. On the other hand, when $d=2$ the Mermin-Wagner theorem applies to the full theory, bulk plus boundary, which is local. In order for the results of the paper to have physical significance, the authors should point out what is the loophole in the theorem which allows the phase transition to exist.
2. The authors cite two papers which study conformal boundary conditions for a $2d$ free boson. In fact, the literature is more extensive than that, and includes a complete classification of the existing boundary conditions, see in particular hep-th/0109021 and references therein. The fixed point at $g=0$, for $N=1$, consists of two copies of a free boson, and in $d=2$ this boundary condition must match one of those discussed in hep-th/0109021 (possibly in the limit of a non-compact boson). Which one? Notice that the boundary spectrum is generically continuous, while in the $\epsilon$ expansion such feature is lost. Like the previous point, this observation calls into question the extrapolation of the results to $\epsilon=1$.
The authors should also address the following minor points:
1. The Introduction and section 2 discuss a different setup than the rest of the paper, since they deal with a boundary potential which preserve the symmetries of the bulk, while in section 3 the bulk symmetries are explicitly broken according to the pattern discussed above. The results of section 2 are used in section 3, but the only comment about this is offered below eq. 23, where the formula is said to hold ''for $N>1$, with $M$ promoted to a matrix''. The authors should clearly write if the formula holds for any symmetry of the potential, or in other words for any matrix $M$. It would also be useful to add a sentence above eq. 4 to explain that in the bulk of the paper the assumption of $O(N)$ symmetry of the potential will be relaxed.
2. In eq. 10, should one of the equal signs be replaced by a minus?
3. The choice of contour in fig. 1 is stated but not justified. Why does the correct prescription include the part of the contour which goes around the branch cut?
4. In the computation of the effective potential in section 2, two contributions are disregarded. The first is $M$ independent and IR divergent, the second is $M$ dependent and UV divergent. It is advisable to extend the explanation on these points. The important pieces of the effective potential are the ones that depend on the classical part of the field. How is this related to the (absence of the) $M$ dependence? And what is the ''suitable subtraction scheme'', i.e. the counterterm, which allows to disregard the UV divergent piece proportional to $\log M$ in eq. 22?
5. There is a typo in the title of section 3, as well as in the first sentence.
6. When describing the model in eq. 24, it is best to also discuss the other relevant and marginal directions which preserve the $O(N)\oplus O(N)$ and $\mathbb{Z}_2$ symmetries. Indeed, all of these couplings will in general be present. The authors should specify that the situation under consideration corresponds to fine tuning the boundary masses of the fields. Furthermore, the boundary term $\partial_\perp \phi^2+\partial_\perp \chi^2$ could also be considered. It is classically marginal: the authors might want to comment on why it is never generated.
7. Figure 2 is very important for the understanding of the paper. Some changes would go a long way in making the conclusions of the paper immediately clear. First of all, the arrows should point towards the IR, rather than the UV. It would also be nice to highlight with background colors and labels the phases and their boundaries. Furthermore, the correspondence between the dots and the fixed points in eq. 28 should also be made more explicit, either in the caption or again with labels. It would also be useful to indicate, perhaps in the caption, what is the order parameter on the line of first order phase transition.
8. The symmetries of a potential are not necessarily revealed by a form like the one in eq. 24, since all the potentials obtained by applying $O(2N)$ rotations describe the same physics. It would be useful to write the potential explicitly in terms of irreps of $O(2N)$, so that the residual symmetry is manifest. See for instance the discussion around 3.11 in 2010.16222.

---

## Round 2 · Referee Report · Anonymous (Referee 3) · 2021-7-4

Report

I understood the reply about the point I raised. It is good to know that this term is canceled when imposing the renormalization conditions. It is still not clear to me how a counter-term proportional to log(M) can actually be allowed, it seems to signal an IR divergence of some sort, but this is probably just a technical nuisance. Since it does not stop the authors from getting a sensible final answer, I am satisfied with the explanation in the edited version. However I realised that there is a typo: "Note that the numerator of the logarithm in (22)" below eq. (22), here "numerator" should be "denominator".

---

## Round 2 · Referee Report · Anonymous (Referee 2) · 2021-7-16

Report

The authors satisfactorily addressed my comments. While I still have some minor doubts about a couple of the answers, I feel they are not important enough to further delay publication. I am therefore happy to recommend the paper for publication.

As a small remark, in footnote 15 "ineducable representations" should probably be "irreducible representations".

---

## Round 2 · Author Response

Dear sci-post editors and referees,

We would like to thank the referees for a very valuable and informative feedback on our work. We have taken your comments into account and wrote an updated version of the paper. We have also answered some specific points of your reviews below.

Referee 1.

We fixed the minor points and added footnote 11 to clarify the subtraction scheme question.

Referee 2.

Main points:

1.We have added a paragraph dedicated to Mermin-Wagner theorem on page 4 and another paragraph on the epsilon->1 limit at the end of section 3 on page 12. While we do agree that in general the physical interpretation of the d=2 case as phase transition is not clear the main features discussed in section 1 such as appearance of Dirichlet mode in the IR should persist (as they do for open strings, boundary sine Gordon model etc.). Furthermore our example in section 3 includes O(N) with N=1 (and it could be that for epsilon=1 it only includes this model), in which case the broken symmetry is discrete and thus allowed by Mermin-Wagner theorem.

  1. We have updated the relevant references and added a remark in conclusions on page 13. Because the non-perturbative extrapolation of epsilon->1 limit is beyond the scope of our paper we can only offer speculative/qualitative remarks at this stage. Nevertheless we believe that even if the compact scalar results apply to our model it is entirely possible that the relevant fixed points actually correspond to Dirichlet conditions in the IR in analogy with the IR fixed point of boundary sine-Gordon models.

Minor points:

  1. We added a clarification above eq 4 and below eq 23.
  2. The equal signs in eq. 10 seem to be in order.
  3. The choice of contour is justified in the text under eq. 16. We added some further explanation in the figure 1 caption.
  4. This is now clarified by footnote 11.
  5. Fixed
  6. We added footnote 14 on page 7 and the sentence around it.
  7. We expanded the figure 2 and its caption.
  8. We added some further discussion under eq. 24 and added eq. 25

Should you require any further clarification, please let us know.

Best regards,

Alexander and Vladimir

---

## Round 2 · List of Changes

1. We extended the paragraph below eq. (22) with footnote 11. It contains a discussion on why we can drop the Log(M)-divergence.

  2. We fixed the typo in the title of section 3.1 and in the first line in subsection 3.1.

  3. We fixed the index typos in eq. (33).

  4. We made the notation of A^x_y appearing in first eq. (34), and made it consistent throughout the paper.

  5. We added a "(phi^2_cl, chi^2_cl)"-bracket for Xi_1 and Xi_2 to highlight that those include terms that are dependent on the terms. I.e. they're not constants which can be ignored.

  6. We have added a paragraph dedicated to Mermin-Wagner theorem on page 4 and another paragraph on the epsilon->1 limit at the end of section 3 on page 12. While we do agree that in general the physical interpretation of the d=2 case as phase transition is not clear the main features discussed in section 1 such as appearance of Dirichlet mode in the IR should persist (as they do for open strings, boundary sine Gordon model etc.). Furthermore our example in section 3 includes O(N) with N=1 (and it could be that for epsilon=1 it only includes this model), in which case the broken symmetry is discrete and thus allowed by Mermin-Wagner theorem.

  7. We added a discussion regarding the 2d extrapolated theory, with the relevant references, in the conclusions on page 13. Because the non-perturbative extrapolation of epsilon->1 limit is beyond the scope of our paper we can only offer speculative/qualitative remarks at this stage. Nevertheless we believe that even if the compact scalar results apply to our model it is entirely possible that the relevant fixed points actually correspond to Dirichlet conditions in the IR in analogy with the IR fixed point of boundary sine-Gordon models.

  8. We clarified above eq. (4) and below eq. (23) that we chose O(N) symmetry groups as an example, but in general we can consider other continuous global symmetry groups.

  9. We justified the choice of contour in the text under eq. (16). We added some further explanation in the figure 1 caption.

  10. We added footnote 14 where we justify why we doesn't consider the relevant operators of normal derivatives of phi^2 and chi^2 on the boundary.

  11. We expanded figure 2 and its caption.

  12. Regarding the symmetries of the model we consider, we expanded the discussion under eq. (24) and added eq. (25) where the symmetry is manifest for the equal flavored case.

---

## Editorial Decision

published